# The Progress in Bioprinting and Its Potential Impact on Health-Related Quality of Life

**DOI:** 10.3390/bioengineering10080910

**Published:** 2023-08-01

**Authors:** Antoniya Yaneva, Dobromira Shopova, Desislava Bakova, Anna Mihaylova, Petya Kasnakova, Maria Hristozova, Maria Semerdjieva

**Affiliations:** 1Department of Medical Informatics, Biostatistics and eLearning, Faculty of Public Health, Medical University, 4000 Plovdiv, Bulgaria; yaneva.antonya@gmail.com; 2Department of Prosthetic Dentistry, Faculty of Dental Medicine, Medical University, 4000 Plovdiv, Bulgaria; 3Department of Healthcare Management, Faculty of Public Health, Medical University, 4000 Plovdiv, Bulgaria; desislavabakova@gmail.com (D.B.); anna.mihaylova@mu-plovdiv.bg (A.M.); petya.kasnakova@mu-plovdiv.bg (P.K.); mariya.v.hristozova@mu-plovdiv.bg (M.H.); mariya.semerdzhieva@mu-plovdiv.bg (M.S.)

**Keywords:** 3D (bio)printing, health-related quality of life, systematic review, personalized medicine, healthcare, bioprinting

## Abstract

The intensive development of technologies related to human health in recent years has caused a real revolution. The transition from conventional medicine to personalized medicine, largely driven by bioprinting, is expected to have a significant positive impact on a patient’s quality of life. This article aims to conduct a systematic review of bioprinting’s potential impact on health-related quality of life. A literature search was conducted in accordance with the Preferred Reporting Items for Systematic Reviews and Meta-Analyses (PRISMA) guidelines. A comprehensive literature search was undertaken using the PubMed, Scopus, Google Scholar, and ScienceDirect databases between 2019 and 2023. We have identified some of the most significant potential benefits of bioprinting to improve the patient’s quality of life: personalized part production; saving millions of lives; reducing rejection risks after transplantation; accelerating the process of skin tissue regeneration; homocellular tissue model generation; precise fabrication process with accurate specifications; and eliminating the need for organs donor, and thus reducing patient waiting time. In addition, these advances in bioprinting have the potential to greatly benefit cancer treatment and other research, offering medical solutions tailored to each individual patient that could increase the patient’s chance of survival and significantly improve their overall well-being. Although some of these advancements are still in the research stage, the encouraging results from scientific studies suggest that they are on the verge of being integrated into personalized patient treatment. The progress in bioprinting has the power to revolutionize medicine and healthcare, promising to have a profound impact on improving the quality of life and potentially transforming the field of medicine and healthcare.

## 1. Introduction

The field of bioprinting is experiencing rapid growth and has made a revolutionary impact on the realms of medical and pharmaceutical sciences, garnering significant attention on a global scale [1]. Bioprinting can be defined as the simultaneous deposition of living cells and biomaterials in a precise layer-by-layer manner, utilizing computer-aided transfer processes to fabricate bioengineered constructs [2].

This technology offers remarkable precision in the spatial arrangement of cells, proteins, DNA, drug particles, growth factors, and biologically active particles, thereby facilitating superior tissue generation and formation. The potential of bioprinting extends to advancing tissue fabrication for the development of physiologically relevant tissue constructs, tissue models, organs, and even organs-on-a-chip models in the fields of medicine and pharmaceutics. The clinical translation of bioprinted tissues and organs, however, faces certain controversies, stemming from the inherent nature of bioprinting itself. The involvement of living cells within the bioink, and the relatively recent utilization of patient-specific cells in bioprinting, have raised valid concerns. Stem cells, such as embryonic stem cells and induced pluripotent stem cells, have emerged as potentially limitless sources of patient-specific cells for the fabrication of tissues and organs [3]. These patient-specific cells can be differentiated and subsequently used for bioprinting or can be bioprinted and then differentiated into multiple lineages, thereby creating tissues and organs with minimal risk of immunogenicity [4].

In recent years, the field of bioprinting has experienced significant advancements and is poised to revolutionize the field of medicine. Experimental research indicates that bioprinting holds the potential to fabricate various biological structures, including skin, blood vessels, cartilage, and even entire organs, such as the heart and liver [5]. Bioprinted models are also employed for testing the efficacy and toxicity of drugs in tumor treatments and other applications [6]. Furthermore, researchers are actively pursuing the ambitious goal of creating regenerative tissues and organs [7,8].

The failure of tissues or organs due to factors like aging, diseases, accidents, and congenital defects poses a critical medical challenge, which is primarily addressed through organ transplantation from either living or deceased donors. However, there is a severe global shortage of human organs available for transplantation, exacerbated by the increasing number of patients in need. Current statistics demonstrate that only a small percentage of individuals awaiting organ transplants are successful [9]. Tissue engineering and regenerative medicine-based therapies are emerging as potential solutions to overcome the shortage of organ donors. Therefore, it is crucial to leverage the expertise of scientists who have been actively engaged in tissue engineering research in recent years. Present-day scientific endeavors and investigations primarily focus on advancing the capabilities of printing organs and tissues [10].

Given demographic trends, the rising prevalence of degenerative diseases, and the growing number of patients, the significance of (bio)printing is increasingly recognized [11]. The ability to create customized implants and original printed models tailored to individual patients highlights the vast potential of this technology, which could greatly enhance patient care and quality of life. The utilization of printed models for pre-operative planning and surgical visualization has already become a standard practice in clinical settings, especially for cases involving complex anatomical deformities [12,13]. Moreover, these printed models could be employed to educate and inform patients about upcoming surgical interventions [5,14].

The biomedical applications of 3D printing are vast and encompass the fabrication of tissues and organs, anatomical models, customized prosthetics and implants, advancements in pharmaceutical drug dosage forms, as well as discoveries and advancements in drug delivery [15,16]. Currently, the emerging field of “3D pharming” presents significant opportunities for the pharmaceutical industry [17]. It involves the fabrication of personalized medications, drug delivery systems, and dosage forms using 3D printing techniques. This innovative approach has the potential to revolutionize drug manufacturing, allowing for customized medications tailored to individual patients’ needs and improving treatment adherence and efficacy [17,18,19]. However, further research and regulatory considerations are necessary to ensure the safety, quality, and regulatory compliance of 3D pharming products before they can be widely implemented in healthcare settings [20].

In recent years, there has been a growing interest in 3D printed conductive materials due to their adequate repeatability, ease of operation, biocompatibility, and high conductivity. These materials have garnered attention for various applications, including engineered tissues and organs, as well as biosensors. The integration of conductive materials in biosensors, such as wearable health-monitoring technology, offers the highly sensitive, accurate, and rapid detection of human metabolites. This technology plays a crucial role in screening, diagnosis, and patient monitoring [21]. With the increasing aging population and technological advancements, the incidence of irreversible tissue injuries is expected to rise. The development of 3D-printed conductive artificial tissues and organs holds the potential to enhance the regeneration process by facilitating improved cell–cell communication and better integration with the natural electrical potential of the human body [22].

Microarray technology has also emerged as a valuable tool in the field of bioprinting [23], allowing for the simultaneous analysis of thousands of genes or genetic variations in a single experiment, enabling researchers and clinicians to assess an individual’s genetic profile and tailor medical treatments accordingly [24]. Microarrays play a crucial role in personalized medicine [25,26,27] and can help identify potential biomarkers for disease diagnosis, prognosis, and treatment selection [28,29,30]. In addition, the integration of genome-wide association studies (GWAS) and microarray could potentially yield a more efficient strategy in comparison to analyzing the datasets separately by enhancing the identification of candidate genes and functions associated with tumor development, providing a more precise comprehension of the underlying mechanisms [31]. By incorporating microarray techniques into the bioprinting process, a deeper understanding of the behavior and functionality of printed tissues and organs can be achieved [32], including optimizing the composition of bioinks, assessing the differentiation status of printed cells, and monitoring the overall quality and functionality of the bioprinted constructs [29,33].

The availability of donor tissues and organs poses a significant challenge in various procedures, particularly in reconstructive and transplantation surgery. The limited supply, morbidity associated with tissue harvest, and potential complications related to immunosuppression hinder the progress in these fields [34,35]. With the emergence of 3D printing software, there is a solution in sight. By extracting digital data from patient images, such as computed tomography, magnetic resonance imaging, or laser scanning, custom-made and personalized constructs can be created for surgical planning and implantation in the future [36]. 

The integration of a biological component has the potential to revolutionize personalized healthcare. This advancement would enable the development of autologous living implants, similar to the patient’s own tissue, thereby transforming the established technology and opening up new horizons in the field [34]. By combining 3D printing with biological materials, personalized and patient-specific solutions can be created, paving the way for enhanced surgical outcomes and personalized treatment approaches. 

An analysis of demographic clinical outcomes, such as morbidity and mortality, is existentially minimal in healthcare research, but a survey on health-related quality of life (HRQoL) is a multi-layered concept that represents the perspective of a patient’s health state, including mental, physical, and emotional functioning [37]. The perception of impact based on HRQoL for each patient, in particular in the use of (bio)printing, is important for improving the patient’s health status and the predictors of treatment success [38]. With the development of additive manufacturing and bioprinting, HRQoL is expected to be increasingly explored in relation to the use of 3D-printed tissues and organs in the personalized treatment of patients [39,40,41,42,43,44].

The application areas of bioprinting are directly related to health-related quality of life. The increasing demands in the medical field, such as the negative changes in demographic factors, the need for organ donors, the shift towards non-animal testing using 3D cell culture platforms, wound care, and joint repair procedures, are inevitable driving advancements in bioprinting. Some of the clinical applications include tissue simulation for drug development and discovery, drug toxicity testing, tissue engineering for regenerative medicine and prosthetic medical devices, and organ transplantation. Various aspects of healthcare, including diagnostics (using medical imaging to create models for visualization), surgical planning, and personalized medicine, can benefit from 3D printing. While the applications of bioprinting may disrupt the current systems of organ and tissue donation, these applications are expected to be further developed in the future compared to other uses of 3D printing. Three-dimensional printing is currently being used or explored in different therapeutic contexts, and as a result, it has the potential to have a significant impact on a wide range of health issues. This undoubtedly has the potential to improve the quality of life for individuals [45].

Currently, there is a discernible growth in the field of transplant medicine worldwide, with the potential to significantly extend the lifespan and enhance the overall quality of life for patients. Successful organ transplants, including cornea, liver, kidneys, lungs, heart, spleen, thyroid gland, parathyroid gland, and pancreas, have been performed, addressing critical medical needs. However, the persistent shortage of organ donors poses a serious challenge [14]. To overcome this issue, it is imperative to explore innovative approaches for creating artificial organs [1]. Three-dimensional bioprinting, as an additive bioproduction technology, offers promising prospects in accelerating translational research and enabling the production of fully functional artificial tissues and organs to replace damaged ones. The advancement of bioprinters and biomaterials is instrumental in shaping the architectural and functional aspects of bioprinting design, contributing to the diverse and evolving landscape of this field [46,47]. 

Significant achievements in the field of bioprinting have paved the way for promising advancements in the replacement of bones, cartilage, blood vessels, and internal organs (such as the heart, kidneys, and liver) in human patients. Ongoing research in reproductive medicine is exploring the application of bioprinting technologies, specifically in the biofabrication of artificial ovaries [8,48,49,50,51]. In the context of bone diseases, the future of 3D bioprinting holds potential for bone regeneration or cartilage printing [52]. However, one of the most promising aspects of bioprinting is the development of a 3D printable bone graft that can be utilized in surgical procedures to aid in the healing process for trauma or resection operations [53,54].

Wound management presents a global challenge, particularly with the increasing prevalence of chronic conditions, like diabetes, obesity, and an aging population. Effective wound healing is hindered by various factors, including underlying illnesses, excessive exudate production, microbial infections, inadequate blood flow, and poor nutrition [15,55,56,57]. 

The impact of 3D printing can be observed in the production of prostheses, which offer notable advantages. These custom-made prostheses are cost-effective, tailored to the individual wearer, and provide enhanced comfort compared to traditional prosthetic devices. Furthermore, they can be manufactured within a shorter timeframe, often within a day [58]. 

Our research focuses on summarizing the methods employed in 3D bioprinting technology and the development of tissue/organ models that have significantly improved health-related quality of life. Various illnesses and injuries, such as heart attacks, strokes, and joint degeneration, can profoundly diminish the quality of life for affected individuals, often resulting in irreparable tissue damage that conventional medical interventions are unable to adequately address [59]. The limited therapeutic efficacy of current treatments primarily revolves around preventing further tissue damage rather than facilitating tissue repair or regeneration [6]. Consequently, patients are left to cope with chronically damaged tissues, leading to diminished quality of life and escalating healthcare costs [60].

The article aims to conduct a systematic review of the potential impact of bioprinting on health-related quality of life. Despite the advancements in bioprinting technology, there is a current lack of comprehensive research on the impact of bioprinting on health-related quality of life. Further investigation is required to assess the outcomes, functionality, and long-term well-being of patients who receive bioprinted organs or tissues. Additionally, studying the economic, social, and patient satisfaction aspects will provide a more holistic understanding of how bioprinting can positively influence health-related quality of life. Bridging this research gap will contribute to harnessing the full potential of bioprinting in improving the overall well-being and quality of life for individuals in need of advanced medical interventions. 

## 2. Materials and Methods

This article presents a systematic review of the potential impact of bioprinting on health-related quality of life. The study follows the guidelines of the Preferred Reporting Items for Systematic Reviews and Meta-Analyses. The materials and methods used in the review are described in accordance with these guidelines [61]. 

### 2.1. Literature Search

The authors conducted a comprehensive literature search on the PubMed, Scopus, Google Scholar, and Science Direct databases. They used various keywords, such as 3D (bio)printing, additive manufacturing, rapid prototyping, health-related quality of life, quality of materials for bioprinting, tissue engineering, opportunities, systematic review, healthcare, and personalized medicine. They also included specific terms related to bioprinting applications, like 3D bioprinted joints, 3D bioprinted cartilage, 3D bioprinted bone, 3D bioprinted vasculature, 3D bioprinted osteochondral implant, 3D bioprinted vascularized bone, 3D bioprinted graft, outcome, and patient satisfaction. 

### 2.2. Eligibility Criteria 

The inclusion criteria for the reviews were as follows: (1) articles published between 2018 and 2023; (2) reviews, systematic reviews, or meta-analyses; and (3) full-text articles. The exclusion criteria included (1) abstracts, (2) short communications, (3) patents and policy makers, (4) case reports, and (5) studies lacking fundamental information about the additive manufacturing process and bioprinting. Language restrictions were not applied, and the results were summarized accordingly. 

### 2.3. Data Analysis

For data analysis, a data extraction form was created using Microsoft Office Excel 2010 to ensure consistency in data extraction and analysis. The selected articles from the databases were organized in an Excel spreadsheet, and duplicate articles were removed. Three authors independently reviewed the abstracts of the articles, and a subset of papers was selected. The full texts of these selected papers were then independently read by the same authors, who made a final selection of the relevant studies. Several sources before 2019 were also a part of this literature view. Due to the relatively new nature of studies on 3D printing and bioprinting applications, there was a limited number of experimental and prospective studies that met all the inclusion and exclusion criteria. After careful selection of the literature, the results of the three authors were compared and discussed until a consensus was reached. 

## 3. Results

The initial search identified 468 potentially relevant articles based on their titles from the four selected databases. After removing duplicates, a total of 253 studies remained. Following the evaluation of abstracts, 96 articles were excluded due to insufficient data or differing study strategies. This left 157 full papers for analysis. Finally, 128 full-text articles were selected for inclusion in this systematic review. Figure 1 provides a PRISMA flow chart illustrating the study selection process.

After a detailed review and discussion of the synthesized scientific literature, we arrived at the most significant potential benefits of bioprinting to improve health-related quality of life. They are presented in Table 1:

## 4. Discussion

### 4.1. Personalised Part Production and Reducing Rejection Risks after Transplantation

The future of 3D bioprinting holds great potential in the field of transplantation, offering significant improvements to the quality of life for individuals worldwide [9]. Tissue engineering plays a crucial role in enabling the production of cell-containing constructs in a controlled manner, eliminating the limitations and inconsistencies associated with manual cell seeding [62]. To achieve successful outcomes, key challenges in biomaterial development, construct design, and system integration need to be addressed. This includes matching degradation to tissue development, providing appropriate mechanical and rheological properties, vascularization of constructs, and creating a sterile and controlled environment that incorporates multiple cells, materials, and manufacturing processes [63,64].

One of the exciting prospects of bioprinting is the ability to print and arrange all the components of a tissue, including cells and matrix materials, in three dimensions to create structures that closely resemble natural tissues [112,113].The precise delivery of living cells with suitable materials in an organized manner, at the right location and in sufficient quantities, within an appropriate environment, is crucial for various emerging technologies [63,65,66].

The development of 3D bioprinting technology allows for the creation of organs that possess structural, mechanical, biological, and metabolic properties similar to those of a normal and healthy organ [11]. This technology has various applications, including tissue engineering scaffolds, cell-based sensors, drug/toxicity screening, and tissue or tumor models [86]. In addition to ink-jet printing and bioplotting, biological laser printing (BioLP) using laser-induced forward transfer has emerged as an alternative method for assembling and patterning biomaterials and cells. The advancement of high-throughput biological laser printers has demonstrated their potential for depositing a wide range of biological components, such as biopolymers, nano-sized particles, and mesenchymal stem cells, which are essential for tissue engineering and regenerative medicine strategies [114]. These findings highlight the important criteria for constructing 3D structures through bioprinting, including writing speed, volume fraction of deposited materials, process resolution, and compatibility with other tissue-engineering methods [54].

Bioprinting technology allows for the production of scaffolds that have a uniform distribution of cells throughout the structure. By strategically placing different cell types within the supportive material, it becomes possible to mimic tissues with multiple cell types or create interfaces between different tissues [7]. The viability and proliferation of printed cells are influenced by the choice of material and design, and different bioprinting techniques have demonstrated varying levels of cell activity after fabrication [67].

In the medical field, 3D printed models have become a valuable tool for surgical planning. They have been utilized in numerous procedures, including full face transplantation, kidney or liver tumor removal, and acetabular reconstructive surgery, among others [12,13,67]. These models have significantly contributed to improving surgical outcomes and enhancing patient care [5,14,115,116,117,118,119,120].

The advancement of stem cell technology, particularly induced pluripotent stem cell (iPSC) technology, combined with advanced bioprinting systems, is expected to revolutionize regenerative medicine and disease modeling. This integration holds the potential to create new models for diseases and tissue constructs, enabling the development and testing of personalized therapies with improved effectiveness and reduced costs [3,4].

Bioprinting is poised to play a crucial role in generating complex, multi-layered arrangements of cells for tissue reconstruction and advancing the field of disease modeling [46]. In the future, personalized tissues and organs can be created (printed) for patients in order to minimize the risk of rejection. This approach will greatly enhance overall organ recovery and tissue regeneration [2].

### 4.2. Increase the Patient’s Chance of Survival

In medical treatments, there is often a need for safe and efficient alternatives, especially when traditional methods and materials have limitations and may not provide optimal benefits to patients [1]. Three-dimensional printing and bioprinting offer promising solutions in such cases [69]. The field of medical engineering has embraced these technologies, finding applications in various specialties, like orthopedics, aesthetic dermatology, cardiovascular, and surgery [70,93,94,121,122].

By utilizing 3D and bioprinting, there is an increased likelihood of successful implantation of mature tissue, leading to improved integration of the implanted tissues with the patient’s native myocardial tissues. This can help restore cardiac function after a cardiac injury. These advancements offer an efficient and safe alternative to traditional methods and materials, increasing the patient’s chances of survival and enhancing overall treatment outcomes [68,71,72,73,74,123].

The high demand for kidney transplantation globally has led to a focus on developing 3D-bioprinted kidneys as a potential solution. Access to bioprinted kidneys could reduce reliance on donor programs and increase the availability of suitable organs for transplantation. Once 3D-bioprinted kidneys become available, it is likely that they will be widely implemented due to the urgent need in the field [124,125].

In the case of burn injuries, the approach to patient care has shifted towards improving functional outcomes rather than just survival. Conventional treatments involve surgical excision of injured skin and reconstruction using skin substitutes. However, these substitutes lack all the necessary skin cell types and fail to replicate native skin physiology [75,76,77,78].

There is a need for continuous research and technological advancements, along with a focus on understanding the epigenetic mechanisms involved in producing new constructs. This will contribute to making 3D bioprinting technology a reliable, efficient, and favorable method for fabricating tissue constructs in the near future.

### 4.3. Reducing Patient Waiting Time

One of the significant benefits of bioprinting is the potential to reduce patient waiting times for organ transplantation. This can lead to more lives being saved and shorter waiting periods for organs [79]. Patients on the organ transplant waitlist can potentially receive custom-fit organs before reaching the final stage of organ failure, improving their chances of survival and quality of life. In addition to organs, bioprinting can also be used to print bones, skin, and tissues, further expanding its impact [80,81,82].

In the field of drug research and development, bioprinting is expected to have a positive impact by reducing the cost and time associated with preclinical discovery. However, the application of bioprinting in this context requires careful consideration of the balance between cost and value, as 3D models can be more complex and expensive compared to traditional 2D models [83,84,85]. Despite these challenges, the potential benefits of bioprinting in drug discovery and development make it an area of ongoing research and exploration.

Three-dimensional bioprinting technology potentially offers high precision and rapid construction, making it suitable for individualized medical treatment and reducing the risk of rejection reactions. However, there are several challenges that need to be addressed for its widespread implementation. These challenges include considerations in biomechanics, material selection for stents, ensuring a bacteria-free environment, achieving the proper molding of printed structures, providing sufficient blood supply to the printed structures, and ensuring their long-term survival. As a result, 3D-bioprinting technology is still undergoing development and requires continuous efforts and breakthroughs from researchers. Currently, its application in clinical settings is limited [86,87].

In the realm of 3D printed implants and prosthetics, the availability of open-source 3D-printing files compatible with various printer-based technologies has significantly increased. These files serve as “blueprints”, from which clinicians can swiftly select and initiate the printing of a 3D product [88]. This decentralization of the manufacturing process eliminates transportation and logistical barriers that often cause treatment delays, granting clinicians broader access to a wide range of print files for their patients. For instance, upon diagnosing severe aortic stenosis, clinicians can promptly download and print the blueprint for a new aortic valve. With the advancements in bioprinting, the valve can even be printed using bioinks, such as collagen, closely resembling native tissue valves, or utilizing the patient’s own valvular cells for a truly personalized product [83,84,86]. Similar processes already exist for dental fillings and neurosurgical cranial plugs [85]. The potential to rapidly download and produce 3D products based on tissue blueprints extends to simple, generic prosthetics as well. The integration of medical images, such as CT scans, with 3D-printing technologies allows for precise customization and anatomical matching [87,126]. Maxillofacial surgery, for example, has extensively utilized 3D printing to generate contour models, guides, splints, and implants, with an average production time of less than 24 h [113]. This technology can be extrapolated to include joint replacements, pacemakers, cochlear implants, and other implantable medical devices. Three-dimensional printing facilitates the incorporation of pharmaceutical agents, such as antimicrobial, immunomodulatory, or analgesic substances, during the printing process, which opens the possibility of creating a new category of bioactive medical implants [34].

### 4.4. Homocellular Tissue Model Generation and Precise Fabrication Process with Accurate Specifications

Orthoregeneration, a field that utilizes 3D bioprinting, aims to restore lost function caused by disease or damage. Advances in bioprinting have made it possible to print tissues such as bone, cartilage, and blood vessels, paving the way for clinical applications. Articular joints, which are responsible for precise movements and mobility, often experience defects that affect individuals from various demographics and pose a significant socioeconomic burden. Arthroplasty procedures, which involve joint replacement, are on the rise, and the number of procedures is expected to triple by 2040 [90]. While orthopedic implants provide benefits, they have limitations in terms of durability and suitability for younger patients, as they cannot grow with the patient. Critical-sized bone defects are of particular concern as they cannot heal naturally within a patient’s lifetime. Addressing these challenges requires further research and development in the field of orthoregeneration [127,128]. The use of 3D-bioprinting technology holds great promise in addressing the limitations associated with critical-sized bone defects and cartilage injuries. These advancements have opened up new avenues for personalized treatments and regenerative medicine [90]. Researchers have focused on developing biocompatible scaffolds for cartilage and bone tissue engineering, driven by the high prevalence of such injuries and the long-term complications they can lead to, such as osteoarthritis. Articular cartilage degeneration is a major concern affecting patients’ quality of life, and engineered cartilage scaffolds have emerged as a potential solution to promote cartilage regeneration. Scaffold-based techniques offer advantages such as reduced complications from donor grafts and shorter recovery times due to improved graft stability. A tissue-engineered 3D scaffold provides a supportive environment for cell attachment, growth, and differentiation [92]. Innovative approaches, such as cold atmospheric plasma treatment combined with drug-loaded nanoparticles integrated into 3D-printed tissue scaffolds, have shown significant potential in promoting cartilage regeneration [129,130].

The study conducted by Patel et al. in 2020 focused on assessing the use of 3D-printed triangular titanium implants (TTI) for sacroiliac joint fusion (SIJF) in patients with SIJ dysfunction [92]. The results of the study demonstrated that SIJF with 3D-printed TTI led to significant improvements in pain, disability, and patients’ quality of life. The effectiveness and safety of the procedure were supported by functional tests and radiographic evaluations. The study reported improvements in physical function and a reduction in the need for pain management medications. These findings highlight the potential of 3D-printed implants in improving patient outcomes and quality of life in the context of sacroiliac joint dysfunction.

### 4.5. Improves Skin Construction Speed and Save Patient Life

The field of 3D bioprinting offers promising applications in saving patients’ lives by improving the speed of skin construction. This technology enables the rapid construction of skin with reduced rejection rates, facilitating easier research and development processes and enabling high-quality research aimed at saving lives [93]. By providing a skin-like structure for transplantation [77], 3D bioprinting reduces the need for donors. The technology deposits skin cells uniformly layer by layer, addressing the challenges posed by complex wound thickness and shape [93,94]. This approach increases the survival rate of patients by effectively addressing damaged areas of the skin. As an alternative to conventional skin grafts, 3D bioprinting has the capacity to create new skin that promotes faster healing but within specific limitations. The ability to mimic and fabricate the largest and most complex multi-layered organ of the body, the skin, using 3D-bioprinting technology, opens up enormous potential for grafts in wound healing, the replacement of burned skin, and the development of in vitro human skin models for product and drug testing [77,94].

Despite various therapeutic regimens available, the clinical assessment and management of chronic wounds remain challenging due to the complex nature of wound healing and long-term treatment requirements. Current approaches, such as cell therapy, gene therapy, growth factor delivery, wound dressings, and skin grafts, have limitations and are not universally effective for all types of wounds, necessitating the development of alternative therapies. Failed wound-healing processes can lead to abnormal scar formation and chronic conditions susceptible to infections. Chronic wounds significantly impact patients’ quality of life, increase morbidity and mortality rates, and impose substantial financial burdens on healthcare systems worldwide. Three-dimensional bioprinting-based strategies, either alone or in combination, can promote faster wound healing and address patients’ needs in terms of effectiveness, cost-effectiveness, and cosmetic outcomes [95,96].

### 4.6. Eliminating/Reducing the Need for Organs Donor

The clinical demand for organ and tissue replacement therapy is increasing steadily [9]. Presently, we are witnessing significant advancements in transplant capabilities worldwide. Organ transplants not only extend patients’ lives but also greatly enhance their quality of life [14]. Various organs, including the cornea, liver, kidneys, lungs, heart, spleen, thyroid gland, parathyroid gland, and pancreas, have already been successfully transplanted [7]. However, the shortage of organ donors poses a serious challenge. Therefore, it is imperative to explore innovative methods for creating artificial organs, which can serve as potential alternatives. These methods allow us to explore and develop optimal implant solutions. Three-dimensional bioprinting, an additive bioproduction technology, has emerged as a promising tool for accelerating translational research. It has the potential to produce artificial tissues and organs that can fully replace damaged organs [97]. This field of research continues to evolve due to the constant advancements in bioprinters and biomaterials, which play crucial roles in shaping the architecture and functionality of bioprinted designs. Given its immense potential, bioprinting has the capacity to eliminate the dependency on organ donors, thereby significantly reducing organ trafficking [9].

### 4.7. Expanding Treatment/Transplantation Possibilities

The treatment of type 1 diabetes (T1D) disease may find a potential solution in the form of a bioprinted bionic pancreas, which addresses many of the limitations associated with current approaches. This innovative method is currently in the development phase [10].

In the context of hair follicle regeneration, in situ bioprinting has demonstrated successful outcomes, with the regeneration of hair follicles and other skin appendages observed after 4 weeks. Importantly, the viability and stemness of the stem cells were only slightly affected by the bioprinting process. These findings highlight the potential of a mechanical engineering approach utilizing in situ bioprinting for hair follicle regeneration, presenting promising prospects for clinical application [131].

Restoring vision has long been a significant objective in ophthalmology. While various approaches, such as 3D bioprinting, stem cell therapy, gene therapy, and implantable devices, have been explored, a definitive solution has yet to be achieved. These methods hold potential for restoring functional vision in visually impaired individuals [3,83,98]. Although minimal vision restoration has been achieved in specific cases using different techniques, the optimization of parameters, such as biocompatibility, spatiotemporal resolution, and cost reduction, is crucial for its widespread utilization [98]. The refractive function of our visual system plays a vital role in accurately focusing ambient light onto the retina’s photoreceptors, enabling us to perceive our surroundings clearly [132]. The cornea, responsible for 70% of the eye’s refractive power, is a crucial component in achieving visual acuity, with its stroma layer contributing 90% of its overall volume [133]. Refractive errors, including astigmatism, myopia (nearsightedness), and hyperopia (farsightedness), are prevalent causes of visual impairment globally and have significant epidemiological and socioeconomic implications recognized by the World Health Organization [134,135].

Traditionally, corrective measures for refractive errors involve the use of glasses or contact lenses, which are generally effective. These methods can be inconvenient and burdensome, impacting an individual’s quality of life and daily activities [136]. Some individuals may find wearing glasses cosmetically undesirable, while contact lenses require regular maintenance, including cleaning and disinfection. Within the realm of 3D bioprinting, a specific technique called digital light processing (DLP) bioprinting utilizes projection technology to polymerize biomimicry inks layer by layer, allowing for the precise production of structures [99]. This process enables the creation of corneal constructs with a smooth surface, which is crucial for maintaining corneal transparency and optical clarity, which is essential for normal vision. According to Jia et al., a notable advantage of DLP-bioprinted photopolymerizable natural hydrogel lenticules is their potential to mitigate long-term complications, such as stromal melt, calcification, and the formation of retroprosthetic membranes that may occur with certain surgical procedures or implantable devices [137].

The development of optimal solutions using innovative techniques holds significant implications for the quality of life, socioeconomic status, and mental health of patients. These techniques, which are currently at various stages of development, ranging from clinical trials to commercial availability, show great promise. Over the past decade, they have led to numerous solutions that have improved the quality of life for individuals affected by various conditions [138].

The production of affordable and functional partial hand prostheses presents challenges due to the individualized nature of amputations. However, a study by Alturkistani et al. in 2020 demonstrated the feasibility of using 3D-design and -printing technologies to create an affordable passive partial hand prosthesis that can be customized for different amputation configurations. Feedback from the patient involved in the study highlighted the device’s compactness, lightweight design, and ease of use as its main advantages. Despite having lower grip strength, the prosthesis fulfilled the patient’s needs by providing a stable grasp and improving their ability to perform bimanual activities. The patient believed that this device would enhance their job performance. In comparison to local cosmetic options or no prosthesis at all, the patient expressed a preference for this functional prosthesis [139].

Esophageal pathologies, such as esophageal atresia, cancers, burns, or stenosis, often necessitate esophageal replacement surgery utilizing portions of the gastrointestinal tract. However, this procedure has its limitations and can give rise to complications, thereby reducing both the quality of life and survival rates of patients. In light of these challenges, esophageal tissue engineering (ETE) has emerged as a promising alternative for repairing defective esophagi. Recent advancements in ETE, particularly in the field of 3D bioprinting, have demonstrated great potential in fabricating well-organized 3D functional structures through the use of cellular and acellular bioinks [100].

### 4.8. Creating More Functional Implants

In the field of dentistry, 3D printing is extensively employed for various procedures, including implant surgery, oral and maxillofacial surgery, orthognathic surgery, endodontics, and prosthodontics. The restoration of aesthetic appearance, particularly in the case of anterior teeth, holds significant potential for revitalizing facial beauty, enhancing speech and chewing abilities, and promoting overall well-being. A recent investigation conducted by Xia, Li, Cai, Shi, Zhao, Jiang, et al. (2018) sheds light on the benefits associated with the utilization of 3D-printed templates and resin composites in the enhancement and restoration of anterior teeth’s aesthetic aspect. This study underscores the convenient, aesthetically pleasing, and functional nature of these techniques. They are regarded as dependable and viable alternatives to conventional approaches for addressing dental caries and repairing fractured teeth [140].

In 2023, Moiduddin et al. conducted a study with the objective of employing polyetheretherketone (PEEK) material and 3D-printing technology to reconstruct zygomatic bone deformities. The primary aim was to achieve precise and aesthetically pleasing restoration of the zygomatic bone while addressing the limitations associated with autogenous bone grafts. The study encompassed an integrated approach involving computer-aided implant design, biomechanical analysis, and the assessment of the fitting accuracy of the PEEK implant. The findings demonstrated the potential of 3D-printed PEEK implants in the replacement of severe zygomatic deformities, thereby significantly improving the overall outcomes for patients. Moreover, the study emphasizes that these customized PEEK implants have the added advantages of reducing surgical time and minimizing hospitalization period due to the absence of donor-site morbidity [141].

The study conducted by Ohara et al. in 2022 aimed to assess patient satisfaction with conventional dentures (CDs) compared to 3D printed digital dentures (DDs). The results indicate that while patient satisfaction with DDs may be slightly lower than that with CDs in terms of phonetics, ease of cleaning, stability, comfort, and overall satisfaction, a significant portion of patients (20%) preferred and utilized DDs fabricated using 3D printing due to reduced social disability and fewer clinic visits [101,142].

In the field of bioprinting, 3D bioprinting offers a high level of control over the patterning of vascular networks during the design and initial construction of constructs. In order to facilitate translation to clinical applications, critical factors, such as long-term tissue viability and biomechanical properties that resemble natural tissues, still need to be definitively demonstrated. It is crucial to develop human-scale vascularized osteochondral constructs with defect-matching geometries that possess integrated material properties capable of promoting repair and regeneration. By adopting this approach, it is possible to eliminate the need for donor-site morbidity and provide patients with personalized reconstructive options, representing a significant advancement in the field of tissue engineering. The clinical potential of 3D-bioprinted implants is immense, as they hold the promise of treating trauma and degenerative joint diseases, thereby benefiting patients worldwide [59,102].

The integration of cells into 3D constructs would have a transformative effect on personalizing pharmaceutical and disease management. In the realm of drug testing, various pharmaceutical companies, including Aspect Biosystems, have been actively developing bioprinted lung tissue since 2015. Additionally, research in 3D bioprinting has expanded to address the management of diseases like type 1 diabetes by printing human beta-like cells capable of insulin secretion in response to glucose. The next significant stride in advancing personalized prosthetics involves replacing defective tissue with tissue-engineered constructs that closely resemble the original tissue [103]. For instance, 3D bioprinting enables the printing of ear constructs with precise shapes matching the missing auricles, utilizing a bioink that supports the formation of new cartilage. By incorporating the patient’s own cartilage cells, an exact match to the contralateral ear’s cartilaginous structure can be achieved. The mastery of 3D bioprinting techniques would merge the accuracy of printed medical prosthetics with the advantages of autologous reconstruction, resulting in replacements that bear an unparalleled resemblance to native tissue [143]. As previously discussed, the ultimate potential of 3D bioprinting lies in the ability to produce patient-specific body parts such as organs and limbs. This has the power to revolutionize personalized medicine and surgery, opening up new frontiers in healthcare [34].

### 4.9. Beneficial for Cancer Treatment

Cancer remains a leading cause of mortality worldwide, causing numerous human deaths and treatment-related trauma. The field of 3D printing has already demonstrated its potential in printing tissues, organs, and cells, and now, it is also making significant contributions to cancer treatment [104]. This technology enables safer and more precise surgeries by allowing surgeons to create 3D models of cancerous parts and analyze them prior to the actual procedure [144]. Utilizing patient-specific models is considered the optimal approach for invasive cancer surgeries [145,146].

Three-dimensional printing assists in predicting post-surgical recovery and mitigates various risks associated with surgery [105]. This innovative technology empowers doctors with improved planning and treatment capabilities, as patient-specific tumor models aid in making more informed treatment decisions [106,107,108]. Three-dimensional-printed tumor models provide valuable insights into a patient’s internal structures and are utilized in the development of personalized anti-cancer drugs [109].

The primary challenge associated with 3D-printing technology in the context of cancer treatment is to develop effective treatment modalities [147]. Scientists and researchers have analyzed the development of tumor meshes using accurate 3D-printed models, which aid in better diagnosing cancer and assessing disease status [110]. This technology has emerged as a valuable tool for cancer-related surgeries, enabling the analysis of tumors and the administration of proper drugs tailored to individual patients. These personalized treatments improve the targeting of cancer cells, thereby enhancing patient outcomes. Moreover, 3D printed surgical guides facilitate chemotherapy procedures, while 3D-printed medical tools and devices assist in precise tumor removal [121,123]. The ability of 3D printing to create complex and customized 3D cell biology holds tremendous potential. In the future, it is anticipated that 3D printing will be utilized to administer appropriate doses for killing cancer cells without causing harm to healthy tissues [104,111].

In various fields, the use of three-dimensional printing has been increasingly adopted due to its capacity for personalization and design. Within the realm of cancer treatment, the standard approach for stages I to III typically involves surgery followed by additional therapies. However, many of the available therapies, such as immunotherapy, chemotherapy, radiation, and hormonal therapy, come with significant side effects that impact patients’ quality of life [148]. A new development in the field involves a 3D-printed, biodegradable implant that responds to laser treatment, offering potential as an adjuvant therapy for cancer by combining chemotherapy and thermal ablation [149]. Additionally, researchers have introduced a three-dimensional porous scaffold fabricated using 3D printing, specifically designed for the treatment of locally recurrent breast cancer and subsequent tissue repair post-surgery. This scaffold utilizes easily accessible materials, demonstrating excellent biocompatibility. Overall, the 3D-printed scaffold shows promise as a viable option for the treatment of locally recurrent breast cancer and the subsequent repair of tissue following surgery [150].

Cancer-associated cachexia (CAC) is a complex syndrome characterized by systemic inflammation, weight loss, and adipose lipolysis, resulting in significant detrimental effects on patients’ quality of life. In fact, CAC directly contributes to the death of 20–30% of cancer patients. The severity of fat loss and remodeling of adipose tissue have been found to be inversely correlated with patient survival outcomes. Understanding the underlying mechanisms of fat loss and identifying potential strategies to prevent this process requires a comprehensive understanding of CAC pathophysiology, particularly within white adipose tissue models. In this study, researchers developed an engineered human white-adipose-tissue (eWAT) model using three-dimensional (3D) bioprinting and induced it with pancreatic cancer cell-conditioned medium (CM) to mimic the conditions observed in CAC. The induction of CM significantly increased lipolysis and extracellular matrix (ECM) accumulation within the 3D eWAT model. To explore the influence of vascularization on lipolysis and CAC progression, the researchers further vascularized the eWATs, as the role of vascularization in this context remained largely unknown. The results revealed that CM induction improved the angiogenesis of the vascularized eWATs (veWATs), which exhibited decreased glycerol release but increased the expression of UCP1 compared to eWATs. The CM contained various unique inflammatory cytokines (e.g., IL-8, CXCL-1, GM-CSF) that likely contributed to eWAT lipolysis, the upregulation of UCP1, and ECM development. Furthermore, in response to CM induction, eWATs secreted inflammatory adipokines associated with cancer metastasis, muscle atrophy, and vascularization (e.g., NGAL, CD54, IGFBP-2). This work demonstrated that the eWAT model serves as a robust tool for investigating cachectic fat loss and the remodeling of adipose tissue, offering valuable insights into CAC physiology and the development of potential therapies [151].

From the literature review, the immense benefit of bioprinting technologies for improving health-related quality of life is unquestionably evident. Providing greater access and more treatment options, as well as offering medical solutions tailored to each individual patient, are of paramount importance for the development of healthcare [89].

Three-dimensional bioprinting offers the opportunity to manage diseases through personalized treatments and to produce therapeutics on an industrial scale. Bioprinting is likely to augment personalized healthcare through the efficient coupling of diagnosis to intervention, translating patient-specific images into tailored implants and prosthetics, advancing cell- and gene-based therapies, and regenerative medicine [34]. Three-dimensional tissues or organoid tumor models derived from individuals can drastically reduce the cost and time of drug discovery, leading to a significant minimization of errors during interventions or side effects (e.g., from drug therapy when patients have comorbidities).

Figure 2 summarizes some of the potential benefits from the progress of bioprinting technology and its effects on the health-related quality of life.

## 5. Conclusions

The application of 3D printing and bioprinting in healthcare shows great potential, although these technologies are still relatively new and require extensive clinical trials and research. Various studies conducted in healthcare specialties, including medicine, dental medicine, and pharmacy, indicate a promising future for 3D and bioprinting applications.

Our findings highlight some of the significant potential benefits of bioprinting in improving the patient’s quality of life. These include personalized part production, saving millions of lives, and reducing rejection risks after transplantation. Bioprinting can also accelerate the process of skin-tissue regeneration, generate homocellular tissue models, and facilitate a precise fabrication process with accurate specifications. Additionally, 3D printing and bioprinting could eliminate the need for organ donors, thereby reducing patient wait time. These advances have the potential to greatly benefit cancer treatment and other research areas, offering medical solutions tailored to each individual patient, and increasing their chances of survival while significantly improving their overall well-being.

However, to advance the field of bioprinting and comprehensively understand its impact on health-related quality of life, additional research is necessary. This includes conducting comparative studies between traditional organ transplantation and bioprinting-based organ transplantation, performing long-term assessments of patients who have received 3D (bio)printed organs or tissues, investigating their immunological response, and conducting clinical trials to evaluate safety and efficacy. Economic evaluation, particularly in terms of quality-adjusted life years (QALY), is also needed to achieve approval and widespread adoption of bioprinting as an efficient and safe alternative.

In conclusion, the paradigm shift from conventional to personalized medicine, which heavily relies on bioprinting, is expected to have a significant positive impact on health-related quality of life. With these technologies, patients may experience reduced pain, improved functionality, and a greater sense of well-being, ultimately contributing to better overall health outcomes.

## Figures and Tables

**Figure 1 bioengineering-10-00910-f001:**
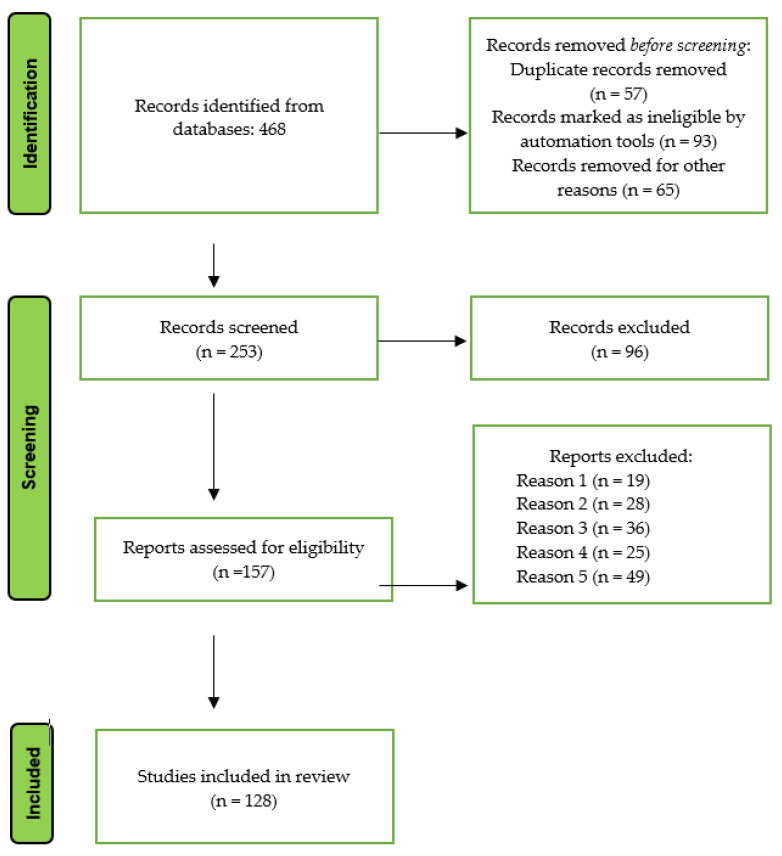
PRISMA flow chart for the selection process of articles.

**Figure 2 bioengineering-10-00910-f002:**
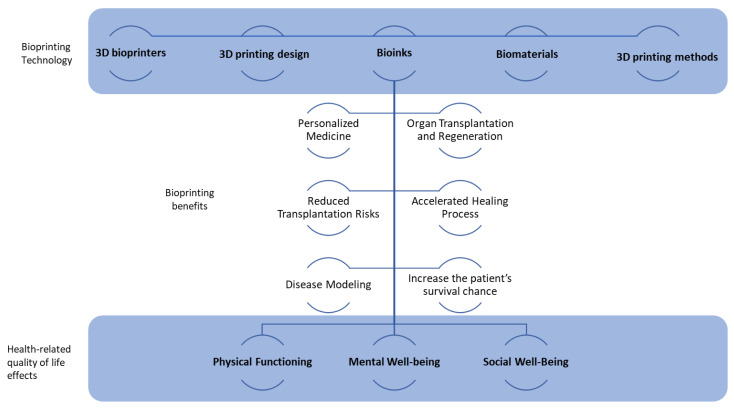
Relationship between bioprinting technology, bioprinting benefits, and health-related quality of life effects.

**Table 1 bioengineering-10-00910-t001:** Potential benefits of bioprinting for health-related quality of life.

Key Benefit/Topic	Area of Application/Significance	References
Personalized part production and reducing rejection risks after transplantation	Production of cell-containing constructs	Gu et al., 2020 [62]
	Biomaterials development	Talebian et al., 2019; Zhang et al., 2021 [63,64]
	Tissue development	Talebian et al., 2019; Zhang et al., 2021 [63,64]
	Ability to print tissue analogue structures	Zhang et al., 2019; Sarmah et al., 2022 [65,66]
	Production of scaffolds with a homogeneous distribution of cells	Veeman et al., 2021; An et al., 2023 [7,67]
	Disease modeling	Shukla et al., 2022; Crook et al., 2020; Fonseca et al., 2020 [3,4,24]
Increase the patient’s chance of survival	3D and bioprinting has great potential to serve as an efficient and safe alternative to the traditional methods and materials	Ahmed et al., 2022 [68]
	3D and bioprinting show promise in becoming effective alternatives	Xu et al., 2019 [69]
	3D printing in the field of medicine through medical engineering	Suvorov et al., 2020 [70]
	3D printing enhances the integration of the implanted tissues	Birla et al., 2020; Shahzadi et al., 2021; Qasim et al., 2019; Roche et al., 2020; Agarwal et al., 2021 [71,72,73,74]
	The focus of 3D printing is to facilitation of improved functional outcomes in patient care	Desanlis et al., 2021; Sedighi et al., 2023; Kamolz et al., 2022; Hann et al., 2019 [75,76,77,78]
Reducing patient wait time	Bioprinting could allow for more lives to be saved and shorter wait times for organs	Gonzalez et al., 2020 [79]
	3D bioprinting decreases the waitlist for organs transplant	Jeong et al., 2020; Barceló et al., 2022; Leberfinger et al., 2019 [80,81,82]
	Bioprinting is expected to reduce the cost and time of preclinical discovery	Ji et al., 2021; Xu et al., 2022; He et al., 2021 [83,84,85]
	3D bioprinting technology has high precision and fast construction speed	Ramos et al., 2020; Condino et al., 2022 [86,87]
	3D bioprinting has potential to generate a new class of bioactive medical implants	Jovic et al., 2020 [34]
	3D bioprinting saves time	Agarwal et al., 2023 [88]
Homocellular tissue model generation and precise fabrication process with accurate specifications	3D bioprinting is able to restore lost function caused by disease or damage	Singh et al., 2019 [89]
	3D bioprinting opens new possibilities for personalized treatments and regenerative medicine	Rosemann et al., 2007 [90]
	Tissue-engineered 3D scaffold would provide the necessary structural support and physical environment for cells to attach, grow, and differentiate	Raghunath et al., 2007 [91]
	3D-printed triangular titanium implants significantly improved pain, disability, and patients quality of life	Patel et al., 2020 [92]
Improves skin construction speed and saves patient life	3D bioprinting holds promising applications to save patient life	Rosca et al., 2020 [93]
	3D bioprinting reduces donor requirements	Kamolz et al., 2022 [77]
	3D-printed skin possesses enormous potential as grafts for wound healing, burned skin replacement, and in vitro human skin models for product and drug testing	Varkey et al., 2019; Kamolz et al., 2022 [77,94]
	3D-bioprinting-based strategies can be used alone or in combination to promote faster wound healing and fulfill patient needs in terms of effectiveness, cost-effectiveness, and cosmetic appearance	Kolimi et al., 2022; Chouhan et al., 2019 [95,96]
Eliminating/reducing the need for organ donors	Production of artificial tissues and organs may completely replace the damaged organ	Yakimova et al., 2020 [97]
	3D bioprinting may eliminate the need for organ donors and reduce organ trafficking	Mills & Mills, 2020 [9]
Offering medical solutions tailored to each individual patient	3D bioprinting, stem cell therapy, gene therapy, implantable devices, etc. have potential to restore functional vision for the visually impaired	Shukla et al., 2022; Ji et al., 2021; Menon et al., 2022 [3,83,98]
	Digital light processing (DLP) bioprinting enables the production of structures with high precision	Wu et al., 2023 [99]
	Using 3D design and printing technologies allow for the creation of customized prosthesis for different amputation configurations	Farhat et al., 2021 [100]
Creating more functional implants	3D implants can reduced the surgical time and hospitalization period due to no donor-site morbidity	Takeda et al., 2020 [101]
	3D bioprinting lends a high degree of control over vascular network patterning during the design and initial building of the construct	Pan et al., 2022; Fang et al., 2019 [59,102]
	3D constructs would transform the ability to personalize pharmaceutical and disease management	Sun et al., 2020 [103]
	3D bioprinting has potential to produce patient-specific body parts, such as organs and limbs, with the capability of revolutionizing personalized medicine and surgery	Jovic et al., 2020 [34]
Beneficial for cancer treatment	3D printing technology helps in the treatment of cancer	Bhuskute et al., 2022 [104]
	3D-printed models eliminates various risk factors during the surgery	Mercader et al., 2019 [105]
	3D-printed patient-specific tumor models can help healthcare professionals make better treatment decisions	Li et al., 2021; Tasneem et al., 2021; Schulze et al., 2021 [106,107,108]
	3D-printed tumor models can used for developing personalized anti-cancer drugs	Pavan & Kumar, 2022 [109]
	3D-printed models also help in improving the diagnosis of cancer treatments	Haleem et al., 2020 [110]
	3D printing will be applied to provide the proper dose to kill cancer cells without damaging the healthy tissues	Bhuskute et al., 2022; Wu et al., 2020 [104,111]

## Data Availability

Not applicable.

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
