# Peer review of "The Progress in Bioprinting and Its Potential Impact on Health-Related Quality of Life"

_bioengineering, 2023, doi:10.3390/bioengineering10080910_

Round 1
Reviewer 1 Report
Comment:
This paper discusses "The progress in bioprinting and its potential impact on 2 health-related quality of life ". The main contribution of the paper is " They have identified some of the most significant potential benefits of bioprinting to improve the patient’s quality of life: personalized part production, saving millions of lives, and reducing rejection risks after transplantation; accelerating the process of skin tissue regeneration; homocellular tissue models generation and precise fabrication process with accurate specifications; eliminating the need for organs donor, and thus reducing patient waiting time. In addition, these advances in bioprinting have the potential to greatly benefit cancer treatment and other research, offering medical solutions tailored to each individual patient that could increase the patient’s chance of survival and significantly improve their overall well-being.."
This is an interesting study and is generally well written and structured. However, in my opinion the paper has some shortcomings in regards to signaling of microarray technology and GWAS. Indeed, short paragraph about those in general will be recommended.
Minor comments:
· Well written except in some situations. I advise recheck it again.
· The introduction should be advised to be re-written to be in more logical flow. And introduce microarray …..
· I prefer to divide the results from discussion
· Please, Suggest future experiments in details
· Please, try to add general paragraph about microarray and discuss it importance personalized medicine.
· There is table. suggest that you make a summary for them as figure.
· Although it needs to be in more logical flow, the introduction provides a good, generalized background of the topic. However, why not cite more literature papers .
· I think the motivations for this study need to be made clearer.
·
· Regarding the figures: I recommend make figures to be illustrative rather than only one table.
· Conclusion is too general/vague. It should be more precise.
Please, rewrite it.
To gain approval and widespread adoption as efficient and safe alternatives, more 648 research in in vivo settings and subsequent clinical trials are necessary to assess their ef- 649 fectiveness and safety in patients. Despite the limitations and the need for further research, 650 current studies demonstrate the practicality of 3D and bioprinting in the medical field. In 651 conclusion, in the near future, 3D and bioprinting could potentially serve as efficient and 652 safe alternatives to traditional methods and materials in the treatment of diseases. How- 653 ever, it will require additional time and research to reach that point.
Author Response
Reviewer 1
Dear Reviewer,
Thank you for your review of our manuscript! We appreciate your positive feedback regarding the main contributions of the paper and its overall structure and writing quality.
Based on your suggestions, we have made the necessary corrections as follows:
Comment:
This paper discusses "The progress in bioprinting and its potential impact on 2 health-related quality of life ". The main contribution of the paper is " They have identified some of the most significant potential benefits of bioprinting to improve the patient’s quality of life: personalized part production, saving millions of lives, and reducing rejection risks after transplantation; accelerating the process of skin tissue regeneration; homocellular tissue models generation and precise fabrication process with accurate specifications; eliminating the need for organs donor, and thus reducing patient waiting time. In addition, these advances in bioprinting have the potential to greatly benefit cancer treatment and other research, offering medical solutions tailored to each individual patient that could increase the patient’s chance of survival and significantly improve their overall well-being."
This is an interesting study and is generally well written and structured. However, in my opinion the paper has some shortcomings in regards to signaling of microarray technology and GWAS. Indeed, short paragraph about those in general will be recommended.
Minor comments:
- Well written except in some situations. I advise recheck it again.
Answer: We appreciate your comment about the overall quality of the manuscript and for giving us the opportunity to recheck the content. We have carefully reviewed it and taken into account all your suggestions. We believe that now our manuscript meets the required standards.
- The introduction should be advised to be re-written to be in more logical flow. And introduce microarray …..
Answer: We have revised the introduction to ensure a smoother and more logical flow, and we have included a short explanation of microarray technology.
Lines: 116-129
- I prefer to divide the results from discussion.
Answer: We have separated the results and discussion sections which indeed enhanced the clarity and organization of the manuscript. Taking your suggestion into consideration, we have restructured the content to have a distinct Results section and Discussion section.
Results section Lines: 251-266
Discussion section Lines: 266-664
- Please, Suggest future experiments in detail.
Answer: Thank you for bringing this to our attention. In the conclusion section, we have added potential future experiments.
Lines: 278-282
- Please, try to add general paragraph about microarray and discuss it importance personalized medicine.
Answer: We have taken your comment into consideration and have incorporated the requested paragraph into the manuscript. Thank you!
Lines: 116-129
- There is table. Suggest that you make a summary for them as figure.
Answer: We have inserted a new figure (figure 2). Thank you!
Lines: 663-666
- Although it needs to be in more logical flow, the introduction provides a good, generalized background of the topic. However, why not cite more literature papers.
Answer: Thank you for your feedback! We appreciate your suggestion to include more literature papers. Based on your recommendation, we have revised the information to include specific citations from relevant studies that support our statements.
Lines: 47-217
- I think the motivations for this study need to be made clearer.
Answer: We have carefully considered your suggestion and have made the necessary revisions to address this concern regarding study motivation. In the revised manuscript, we have more clearly described the research gap in the existing scientific literature, which our study aims to address.
Lines: 209-217
- Regarding the figures: I recommend make figures to be illustrative rather than only one table.
Answer: We appreciate your suggestion to make the figures more illustrative rather than relying solely on a table. We have made the necessary changes to improve the visual representation of our data and we have inserted a new figure (figure 2).
Lines: 663-666
Conclusion is too general/vague. It should be more precise.
Answer: Thank you for your comment regarding the conclusion and we agree that it should be more precise. In the revised version, we have reworked the conclusion as we have highlighted the key findings and their implications, clearly outlining the contributions and significance of our research.
Lines: 667-692
Comments on the Quality of English Language
Please, rewrite it.
To gain approval and widespread adoption as efficient and safe alternatives, more research in in vivo settings and subsequent clinical trials are necessary to assess their effectiveness and safety in patients. Despite the limitations and the need for further research, current studies demonstrate the practicality of 3D and bioprinting in the medical field. In conclusion, in the near future, 3D and bioprinting could potentially serve as efficient and safe alternatives to traditional methods and materials in the treatment of diseases. However, it will require additional time and research to reach that point.
Answer: Thank you for your feedback! We have taken your suggestion into consideration and have rewritten the paragraph accordingly. Please find it in the revised version of the conclusion.
Lines: 670-695
Reviewer 2 Report
In this study, authors have present the 3D bioprinting applications, significance and its impacts on quality of life. However, authors must address the issues:
· In line 95, “3D pharming” as a 3D printing technology for tailored pharmaceutical drugs should be mentioned. Authors should cited the related article.
· Spirtam as a first FDA approved pharmaceutical dosage form with benefits in healthcare should be mentioned.
· The limitation of some 3D printing techniques like SLA, SLS, FDM and … for healthcare engineering should be stated.
· For 3 sequential paragraphs [Lines 109, 113, 121], words with the same meaning (Furthermore, In addition, Moreover) have been repeated for staring. It must revise.
· Application of 3D printing in some tissue engineering can be expanded to different tissues like bone regeneration. In this regard some articles like https://doi.org/10.1016/j.bioadv.2022.213254 can be cited.
· Keywords of “additive manufacturing” and “rapid prototyping” should be added to literature search on the given database.
· In table 1, area of application/significance, fabrication of anatomical model for education is another item.
· 4D bioprinting must be included in an appropriate section of the manuscript.
· More future perspectives and current limitations must be mentioned in the manuscript.
Best
Minor edition is necessary.
Author Response
Reviewer 2
Dear Reviewer,
Thank you for reviewing our study! We appreciate your valuable feedback and the specific issues you have raised.
Based on your suggestions, we have made the necessary corrections as follows:
In this study, authors have presented the 3D bioprinting applications, significance and its impacts on quality of life. However, authors must address the issues:
- In line 95, “3D pharming” as a 3D printing technology for tailored pharmaceutical drugs should be mentioned. Authors should cited the related article.
- Spirtam as a first FDA approved pharmaceutical dosage form with benefits in healthcare should be mentioned.
Answer: We appreciate your comments and suggestions. We agree that the topic of "3D pharming" is relevant to our study on 3D bioprinting applications and their impact on the quality of life. We have made the necessary revisions to address this issue. In our revised manuscript, we have included the term "3D pharming" as one of the applications of 3D bioprinting technology for the production of personalized pharmaceutical drugs. Furthermore, we have cited the related article that you kindly suggested.
Lines: 94-104
- The limitation of some 3D printing techniques like SLA, SLS, FDM and … for healthcare engineering should be stated.
Answer: We appreciate your suggestion to study and compare specific 3D printing techniques, but it is important to note that the primary purpose of our manuscript is to evaluate the impact of bioprinting technology as a whole on health-related quality of life and does not involve such analyses. However, we value your feedback and will certainly keep it in mind for our future research related to the topic.
- For 3 sequential paragraphs [Lines 109, 113, 121], words with the same meaning (Furthermore, In addition, Moreover) have been repeated for staring. It must revise.
Answer: We have revised the manuscript to address this issue. Instead of using "Furthermore," "In addition," and "Moreover" successively, we have now utilized alternative phrases to avoid repetitive language.
- Application of 3D printing in some tissue engineering can be expanded to different tissues like bone regeneration. In this regard some articles like https://doi.org/10.1016/j.bioadv.2022.213254 can be cited.
Answer: We have revised the manuscript and included the appropriate citation to the article you mentioned. Thank you!
Lines:180-188
- Keywords of “additive manufacturing” and “rapid prototyping” should be added to literature search on the given database.
Answer: In the revised version, we have carefully selected and included appropriate keywords that accurately reflect the content of our study.
Line: 226
- In table 1, area of application/significance, fabrication of anatomical model for education is another item.
Answer: We appreciate your suggestion and the importance of the fabrication of an anatomical model for education, but this aspect does not directly impact health-related quality of life (HRQL). We will certainly take your suggestion into account in future research.
- 4D bioprinting must be included in an appropriate section of the manuscript.
Answer: Given the fact that it is a relatively new technology, 4D bioprinting is not yet associated with potential benefits of its application with HRQL. Thank you for the proposal, which we intend to develop in our future publications.
- More future perspectives and current limitations must be mentioned in the manuscript.
Answer: Thank you! We appreciate your comment regarding the future perspectives and current limitations of bioprinting. In the revised version, we have reworked the conclusion to provide more details of the key findings, future perspectives, and limitations.
Lines: 670-695
Comments on the Quality of English Language
Minor edition is necessary.
Reviewer 3 Report
In their review manuscript, Yaneva et al. have summarized potential benefits of using bioprinting in health care to improve the quality of patients’ life. Although the topic is time-relevant, the main problem of the manuscript is its lack of new information. Compared to the recent review from the authors on the same topic ((Bio)printing in Personalized Medicine—Opportunities and Potential Benefits), the current manuscript is clearly less informative and more loosely written. Often, the authors repeat loose sentences about the tremendous or vast potential of bioprinting without providing any further insight into the topic. Therefore, the review feels like a long, rather superficial list of potential uses of bioprinting, lacking thorough discussion of the feasibility or challenges of these applications. Furthermore, the text discusses repetitively the same applications, such as skin or organ transplants, in different parts of the text, impairing the logic of the manuscript. Also, related to this, the introduction is unnecessarily long, as it goes through the same topics as the results section. Use of an illustration along with the text could make the logic more understandable.
As examples of the loose sentences, the following sentences would benefit from more precise wording and further details:
“3D bioprinting technology offers high precision and fast construction speed, making it suitable for individualized medical treatment and reducing the risk of rejection reactions.”
- How exactly do the high precision and fast construction speed of bioprinting reduce the risk of rejections?
“This technology {bioprinting} enables the rapid construction of skin with reduced rejection rates, facilitating easier research and development processes and enabling high-quality research aimed at saving lives.”
- Again, how does the bioprinting reduces the rejection rates? And how does it enable high-quality research? Quite often, the research done with the help of bioprinting is rather mediocre instead of high-quality.
“By utilizing 3D and bioprinting, there is an increased likelihood of successful implantation of mature tissue, leading to improved integration of the implanted tissues with the patient's native myocardial tissues.”
- How does bioprinting increase the likelihood of successful implantation of tissue? Why does the bioprinted tissue integrate better with the native tissue compared to non-printed tissue?
“In the case of burn injuries, the approach to patient care has shifted towards improving functional outcomes rather than just survival. Conventional treatments involve surgical excision of injured skin and reconstruction using skin substitutes. However, these substitutes lack all the necessary skin cell types and fail to replicate native skin physiology.”
- Burn injuries can be currently treated with lab grown skin tissue that has native skin cell types and skin physiology. How is bioprinting improving the current situation?
“One of the significant benefits of bioprinting is the potential to reduce patient waiting times for organ transplantation.”
- How long are the waiting times now and how long would it be if bioprinting is used? What is needed before the bioprinted organs are realistic?
“While orthopedic implants provide benefits, they have limitations in terms of durability and suitability for younger patients, as they cannot grow with the patient. … The use of 3D bioprinting technology holds great promise in addressing the limitations associated with critical-sized bone defects and cartilage injuries.”
- How can bioprinting improve the suitability of orthopedic implants for younger, growing patients?
In their conclusions, the authors claim that “many unanswered questions remain, and thorough research is needed to ensure the proper implementation of 3D printing in medical practice”.
- What are these unanswered questions and what kind of research is exactly needed? Which breakthroughs would make bioprinting more useful in health care? As this is an important topic, it could be discussed in its own chapter.
Moreover, in the following text, the numbers do not make sense and also do not match the numbers in Figure 1:
“The initial search identified 468 potentially relevant articles based on their titles from the four selected databases. After removing duplicates, a total of 923 studies remained. Following the evaluation of abstracts, 96 articles were excluded due to insufficient data or differing study strategies. This left 108 full papers for analysis.“
The language could be more precise, as described above.
Author Response
Reviewer 3
Dear Reviewer,
Thank you for your comments! We appreciate your feedback and the opportunity to address the concerns raised. We have carefully revised and updated the manuscript according to your suggestions. Additionally, we have taken great care to ensure that the content is presented in a coherent and logical manner, eliminating unnecessary repetition. In response to your comment about the excessive length and redundancy of the introduction, we have thoroughly streamlined it to provide a concise overview while emphasizing the unique contributions of our work. We have also included appropriate illustrations to enhance the clarity and coherence of the manuscript, aiding in the understanding of the topics discussed.
We thank you for your valuable feedback, which has greatly contributed to the improvement of our manuscript.
Based on your suggestion, we have made the necessary corrections as follows:
Comment:
In their review manuscript, Yaneva et al. have summarized potential benefits of using bioprinting in health care to improve the quality of patients’ life. Although the topic is time-relevant, the main problem of the manuscript is its lack of new information. Compared to the recent review from the authors on the same topic ((Bio)printing in Personalized Medicine—Opportunities and Potential Benefits), the current manuscript is clearly less informative and more loosely written. Often, the authors repeat loose sentences about the tremendous or vast potential of bioprinting without providing any further insight into the topic. Therefore, the review feels like a long, rather superficial list of potential uses of bioprinting, lacking thorough discussion of the feasibility or challenges of these applications. Furthermore, the text discusses repetitively the same applications, such as skin or organ transplants, in different parts of the text, impairing the logic of the manuscript. Also, related to this, the introduction is unnecessarily long, as it goes through the same topics as the results section. Use of an illustration along with the text could make the logic more understandable.
As examples of the loose sentences, the following sentences would benefit from more precise wording and further details:
“3D bioprinting technology offers high precision and fast construction speed, making it suitable for individualized medical treatment and reducing the risk of rejection reactions.”
- How exactly do the high precision and fast construction speed of bioprinting reduce the risk of rejections?
Answer: Thank you! We appreciate your feedback. The high precision and fast construction speed of 3D bioprinting reduce the risk of rejection reactions by enabling the creation of customized structures that closely resemble the patient's own tissues, providing better compatibility and integration, minimizing the use of foreign materials, and expediting the transplantation process. However further research is necessary to fully understand the relationship between bioprinting parameters, immune response, and rejection reactions. We think that continued research and collaboration between multidisciplinary fields will play a crucial role in uncovering the full potential of bioprinting for the benefit of health-related quality of life.
“This technology {bioprinting} enables the rapid construction of skin with reduced rejection rates, facilitating easier research and development processes and enabling high-quality research aimed at saving lives.”
- Again, how does the bioprinting reduces the rejection rates? And how does it enable high-quality research? Quite often, the research done with the help of bioprinting is rather mediocre instead of high-quality.
Answer: Our claims are based on scientific data for a potential reduction of rejection rates in various experimental studies, some of them we have cited in our manuscript:
- Gu, Z.; Fu, J.; Lin, H.; He, Y. Development of 3D bioprinting: From printing methods to biomedical applications. Asian Journal of Pharmaceutical Sciences 2020, 15, 529–557. 10.1016/J.AJPS.2019.11.003.
- Talebian, S.; Mehrali, M.; Taebnia, N.; Pennisi, C.P.; Kadumudi, F.B.; Foroughi, J.; Hasany, M.; Nikkhah, M.; Akbari, M.; Orive, G.; et al. Self-Healing Hydrogels: The Next Paradigm Shift in Tissue Engineering? Advanced Science 2019, 6, 1801664. 10.1002/ADVS.201801664.
- Zhang, J.; Wehrle, E.; Rubert, M.; Müller, R. 3D Bioprinting of Human Tissues: Biofabrication, Bioinks, and Bioreactors. International journal of molecular sciences 2021, 22. 10.3390/IJMS22083971.
- Zhang, B.; Gao, L.; Ma, L.; Luo, Y.; Yang, H.; Cui, Z. 3D Bioprinting: A Novel Avenue for Manufacturing Tissues and Organs. Engineering 2019, 5, 777–794. 10.1016/J.ENG.2019.03.009.
- Sarmah, J.K.; Dutta, A.; Sarmah, S.; Ankaleswar, B. Guar gum nanoparticles: A new paradigm in biomedical applications. Polysaccharide Nanoparticles: Preparation and Biomedical Applications 2022, 119–144. 10.1016/B978-0-12-822351-2.00003-6.
- Veeman, D.; Sai, M.S.; Sureshkumar, P.; Jagadeesha, T.; Natrayan, L.; Ravichandran, M.; Mammo, W.D. Additive Manufacturing of Biopolymers for Tissue Engineering and Regenerative Medicine: An Overview, Potential Applications, Advancements, and Trends. International Journal of Polymer Science 2021, 2021. 10.1155/2021/4907027.
- An, C.; Zhou, R.; Zhang, H.; Zhang, Y.; Liu, W.; Liu, J.; Bao, B.; Sun, K.; Ren, C.; Zhang, Y.; et al. Microfluidic-templated cell-laden microgels fabricated using phototriggered imine-crosslinking as injectable and adaptable granular gels for bone regeneration. Acta biomaterialia 2023, 157, 91–107. 10.1016/J.ACTBIO.2022.11.034.
“By utilizing 3D and bioprinting, there is an increased likelihood of successful implantation of mature tissue, leading to improved integration of the implanted tissues with the patient's native myocardial tissues.”
- How does bioprinting increase the likelihood of successful implantation of tissue? Why does the bioprinted tissue integrate better with the native tissue compared to non-printed tissue?
Answer: We do not deny the implantation techniques used so far. The goal of our scientific research is to present the additional possibilities of 3D printing to improve treatment outcomes and quality of life. The provision of native tissue can in some cases have a negative impact on patients and poses risks due to additional interventions (eg bypass placement).
“In the case of burn injuries, the approach to patient care has shifted towards improving functional outcomes rather than just survival. Conventional treatments involve surgical excision of injured skin and reconstruction using skin substitutes. However, these substitutes lack all the necessary skin cell types and fail to replicate native skin physiology.”
- Burn injuries can be currently treated with lab grown skin tissue that has native skin cell types and skin physiology. How is bioprinting improving the current situation?
Answer: According to the studied scientific literature, we believe that bioprinting technologies expand the possibilities of dealing with burn injuries skin reconstruction.
“One of the significant benefits of bioprinting is the potential to reduce patient waiting times for organ transplantation.”
- How long are the waiting times now and how long would it be if bioprinting is used? What is needed before the bioprinted organs are realistic?
Answer: Thank you for your comments and suggestions! The lack of organs has reached a critical stage, with a significant imbalance between the number of patients on the waiting list and the availability of donors. In 2020, the Global Donation and Transplantation Observatory reported a decrease of 17.6% in the number of organ transplants worldwide, with 129,681 transplants compared to 157,301 in 2019. Tragically, statistics indicate that approximately 20 patients lose their lives each day while awaiting organ transplant opportunities. In this context, 3D printing holds promise for transplantation and has the potential to significantly enhance the quality of life for numerous people worldwide.
“While orthopedic implants provide benefits, they have limitations in terms of durability and suitability for younger patients, as they cannot grow with the patient. … The use of 3D bioprinting technology holds great promise in addressing the limitations associated with critical-sized bone defects and cartilage injuries.”
- How can bioprinting improve the suitability of orthopedic implants for younger, growing patients?
Answer: We have discussed this topic in our article “(Bio)printing in Personalized Medicine—Opportunities and Potential Benefits”:
“…As a future goal, the thesis of regenerating organs and organs that grow with the growth of the organism is also being developed, which is especially important in neonatology [12,13]…”
In their conclusions, the authors claim that “many unanswered questions remain, and thorough research is needed to ensure the proper implementation of 3D printing in medical practice”.
- What are these unanswered questions and what kind of research is exactly needed? Which breakthroughs would make bioprinting more useful in health care? As this is an important topic, it could be discussed in its own chapter.
Answer: Health-related quality of life (HRQoL) research can be used in a wide variety of applications. One of the most important applications is for estimating the economic medical and health care interventions. The three most widely used methods for economic evaluation are cost-benefit analysis (CBA), cost-effectiveness analysis (CEA), and cost-utility analysis (CUA). The theoretical basis and conclusions that can be drawn from each can differ. HRQoL refers to patient-reported outcomes of functioning and well-being in physical, mental, and social domains of life. Some of the most commonly used validated instruments are the SF-36, EQ-5D, Health Utilities Index (HUI), self-administered Quality of Well-being Scale (QWB-SA), and the Health and Activities Limitation Index (HALex) questionnaires.
Now, there is a lack of literature data on conducted research on comprehensive measurement of health outcomes includes a combination of life expectancy and health-related quality of life related to the results of the application of 3D printing in health care. Since it also concerns a subjective assessment, this implies the emergence of possible questions related to the subjective feelings/judgment of the patients, which at the moment we cannot comment on. Thank you for your suggestion! We would gladly direct our attention to this important topic in our future research work.
Moreover, in the following text, the numbers do not make sense and also do not match the numbers in Figure 1:
“The initial search identified 468 potentially relevant articles based on their titles from the four selected databases. After removing duplicates, a total of 923 studies remained. Following the evaluation of abstracts, 96 articles were excluded due to insufficient data or differing study strategies. This left 108 full papers for analysis. “
Answer: Thank you for pointing out the inconsistency in the numbers presented in the text and Figure 1! We apologize for the confusion caused. It was indeed a mistake on our part, and we have already corrected it in the revised version of the manuscript.
The corrected statement should read as follows:
The initial search identified 468 potentially relevant articles based on their titles from the four selected databases. After removing duplicates, a total of 253 studies remained. Following the evaluation of abstracts, 96 articles were excluded due to insufficient data or differing study strategies. This left 157 full papers for analysis. Finally, 128 full-text articles were selected for inclusion in this systematic review. Figure 1 provides a PRISMA flow chart illustrating the study selection process.
Lines:252-257
Comments on the Quality of English Language
The language could be more precise, as described above.